# Intracellular Mis-Localization of Modified RNA Molecules and Non-Coding RNAs: Facts from Hematologic Malignancies

**DOI:** 10.3390/cimb47090758

**Published:** 2025-09-14

**Authors:** Argiris Symeonidis, Argyri Chroni, Irene Dereki, Dionysios Chartoumpekis, Argyro Sgourou

**Affiliations:** 1School of Health Sciences, Faculty of Medicine, Hematology Division, University of Patras, 26504 Patras, Greece; argychroni@gmail.com; 2Biology Laboratory, School of Science and Technology, Hellenic Open University, 26335 Patras, Greece; ntereki.eirini@ac.eap.gr; 3Department of Internal Medicine, Division of Endocrinology, Medical School, University of Patras, 26335 Patras, Greece; dchart@upatras.gr

**Keywords:** non-coding RNAs, intracellular topography, hematological malignancies, RNA modification, abnormal signalling

## Abstract

The intracellular topography of RNA molecules, encompassing ribonucleotides with biochemical modifications, such as N6-methyladenosine (m6A), 5-methylcytosine (m5C), adenosine to inosine (A → I) editing, and isomerization of uridine to pseudouridine (Ψ), as well as of non-coding RNA molecules, is currently studied within the frame of the epigenome. Circulating RNA molecules in the intracellular space that have incorporated information by carrying specific modifications depend on the balanced activity and correct subcellular installation of their modifying enzymes, the “writers”, “readers” and “erasers”. Modifications are critical for RNA translocation from the nucleus to the cytoplasm, for stability and translation efficiency, and for other, still-uncovered functions. Moreover, trafficking of non-coding RNA molecules depends on membrane transporters capable of recognizing signal sequences and RNA recognition-binding proteins that can facilitate their transport to different intracellular locations, guiding the establishment of interconnection possibilities with different macromolecular networks. The potential of long non-coding RNAs to form multilayer molecular connections, as well as the differential topology of micro-RNAs in cell nuclei, compared to cytoplasm, has been recognized by several studies. The study of the intercellular compartmentalization of these molecules has recently become feasible thanks to technological progress; however, a wealth of information has not yet been produced that would lead to safe conclusions regarding non-coding RNA’s contributions to the early steps of pathogenesis and disease progression in hematological malignancies. Both, the bone marrow, as the main hematopoietic tissue, and the lymphoid tissues are composed of cells with highly reactive potential to signals affecting the epigenome and initiating cascade pathways in response. Independently or in combination with coexistent driver genetic mutations, especially mutations of enzymes involved in epigenomic surveillance, intracellular microenvironmental alterations within the cell nuclear, cytoplasmic, and mitochondrial compartments can lead to disorganization of hematopoietic stem cells’ epigenomes, promoting the generation of hematological malignancies. In this review, we discuss the various intracellular processes that, when disrupted, may result in the ectopic placement of RNA molecules, either inducing specific modifications or non-coding molecules or promoting hematological malignant phenotypes. The crosstalk between mitochondrial and nuclear genomes and the complex regulatory effects of mis-localized RNA molecules are highlighted. This research approach may constitute a field for new, more specifically targeted therapies in hematology based on RNA technology.

## 1. Introduction

A variety of covalent bonded modifications are mainly prevalent, following protein and RNA, rather than DNA, molecule biosynthesis. These chemical modifications represent an efficient way of post-transcriptional/post-translational regulation of the function of macromolecules and of an additional control of gene expression profiles, providing a high degree of adaptation to the continuously changing cellular micro- and macro-environment [1]. Chemical modification of the macromolecules has been extensively highlighted and studied over the last decades, since Conrad Waddington [2,3] (1905–1975) introduced the term “epigenetic mechanisms” to describe it. Thereafter, epigenetic signatures have gradually been uncovered and have revealed their potential as disease-causing and disease-modifying mechanisms [4]. The field of RNA epitranscriptomics has been delayed from being thoroughly studied due to lack of highly sensitive quantitative and qualitative detection techniques. Although RNA modifications were described over 50 years ago, only recent studies have uncovered that their presence and binding partners are dysregulated in cancer, creating unique opportunities for new therapeutic approaches [5,6]. Currently, more than 170 post-transcriptional modifications have been identified in cellular RNA, apart from the 5′ cap and 3′ poly-A tail, and more than 350 proteins involved in these modifications have been included in the MODOMICS database [7]. Modifications can occur via the addition of simple or complex chemical groups to the ribonucleosides. Modifications, other than chemically-induced, have also been detected, such as isomerization (e.g., uridine to pseudouridine [Ψ]), oxidation (e.g., 5-methylcytosine to 5-methylcytidine and 5-hydroxymethylcytosine to 5-hydroxymethylcytidine), reduction (e.g., uridine to dihydrouridine), and substitution (e.g., uridine to 4-thiouridine). They are spread among all kinds of RNA and are apparent in nuclear, cytoplasmic, and mitochondrial RNAs [7,8]. RNA modifications and the associated modifying enzymes or binding proteins are known to be implicated in homeostasis of HSCs, providing quiescence balance self-renewal and differentiation capacity, and consequently, instructing cell fate decisions. When RNA modifications are disrupted, the situation appears to favor the development of clonal hematologic malignancies. Alterations in RNA splicing and RNA editing during HSC aging due to defective RNA modification pathways contribute to increased myeloid lineage skewing and activation of inflammation-responsive transcription factors [9]. These facts underscore the importance of epitranscriptomic mechanisms in the acquisition of HSC age-related abnormal phenotypes.

Additionally, non-coding (nc) RNAs represent another group οf epigenetic players which are strongly associated with the development of clonal/neoplastic hematological disorders. Here, we provide an in-depth overview of how subcellular mis-localization of ncRNAs, mainly microRNAs (miRs) and long non-coding RNAs (lncRNAs), may stimulate their differentiated activity and their participation in alternative intra-molecular roots, as well as how these mechanisms might contribute to the generation of hematological malignancies.

The precise subcellular localization of all the macromolecules is crucial for the integrity and specificity of their function. Altered localization, caused by specific mutation profiles, stress signals, or structural nuclear defects, can modulate their regulatory roles. The most profound and well-studied paradigm in hematology is the AML subtype, which carries mutations in the nucleophosmin1 gene (*NPM1*) that direct nucleophosmin1 molecule to translocate in the cytoplasm, possibly misrouting other molecules and altering various gene regulatory networks in both compartments [10]. Under physiological conditions, NPM1 phosphoprotein shuttles between the nucleus and cytoplasm, chaperoning ribosomal proteins and core histones between subcellular compartments, but *NPM1* mutations trigger its favorable localization in the cytoplasmic cell compartment [11,12,13]. Other examples implicating ncRNAs, such as hsa-miR-223, hsa-miR-690 [14,15,16], lncRNA-HOTAIRM1, lncRNA-HOTTIP, etc. [17,18,19], which are discussed in detail in separate sections of this review, provide clear evidence of how ncRNA mis-localization directly and indirectly contributes to disease pathogenesis.

NcRNAs in the extracellular matrix constitute mostly signaling molecules transported to other areas of the same or different tissues and are not within the purposes of this review. On the contrary, we address issues raised by the altered compartmentalization of intracellular ncRNA molecules and defective pathways related to m6A, m5C, A → I and Ψ modified RNAs and to the underlying mechanisms, discussing also the potential effects on cells’ fate within the hematopoietic compartment [20,21,22,23,24,25,26,27,28]. The composition of post-transcriptional modifications in RNA transcripts can determine their decay versus stabilization/increased translation; due to the subcellular localization of associated RNA-modifying enzymes, they can potentially impinge different aspects of transcript metabolism. Finally, modified expression/activity of ‘reader’ proteins can serve as the final arbiters of the fate of specifically decorated transcripts [29,30,31,32,33,34]. We speculate that these functional variabilities may, at least in part, explain some experimental observations combining altered RNA modification and topology with tumorigenesis in hematology.

## 2. Subcellular Dynamics of Chemically-Modified RNA Molecules and Their Modifying Enzymes in Hematologic Malignancies

N6-methyladenosine (m6A) is one of the most abundant RNA modifications in mammalian cells. Additional well-documented marks in RNA molecules include 5-methylcytosine (m5C), the conversion of adenosines to inosines (A → I) in double-stranded RNA (dsRNA) substrates, and the isomerization of uridine to pseudouridine, which have been reported to be associated with hematologic malignancies [20]. 2′-O-Methylation is only briefly mentioned, since there are very few references implicating the alternative topology of RNA molecules carrying this modification in hematologic malignancies [35,36] (Figure 1A).

### 2.1. N6-Methyladenosine RNA Modification

m6A is a methylation modification located on the sixth nitrogen atom of RNA adenine and is the most prevalent and well-studied internal modification in eukaryotic mRNA and ncRNAs, affecting various biological processes, including chromatin conformation and consequent regulation of gene expression, RNA processing and half-life, translation, etc. [21,37,38,39]. m6A occurs both in the nuclear and cytoplasmic cell compartments and is mainly distributed in the internal gene exons and the 3′ untranslated regions, with a significant enrichment just upstream of the stop codon of messenger RNAs and of long non-coding RNAs [22].

In the nucleus, m6A is primarily added to immature RNA molecules during transcription, acting as a functional signal for downstream mRNA maturation. The m6A mark, as well as the modifying enzymes associated with m6A RNA, is implicated in maturation processes, including splicing (and alternative splicing) [29], function of the spliceosome component U6 [30], RNA export to the cytoplasm through specified transporters, and RNA stability during hydrolysis [23]. Among ncRNA species, m6A methylation of micro-RNA (miRNA) primary transcripts has been documented as a requirement for their efficient recognition and enzymatic processing to mature miRNA molecules [40].

In the cytoplasm, m6A modification influences translation efficiency and mRNA decay [39,41]. Many studies support that m6A contributes to surveillance pathways to selectively degrade faulty transcripts, by enhancing and sensitizing the nonsense-mediated mRNA decay (NMD) clearance from cells [42]. NMD is a quality control mechanism that selectively degrades mRNAs containing premature stop codons (nonsense mutations) to prevent the production of truncated, potentially harmful proteins. The m6A mark recruits UPF1, a critical NMD factor, to enhance the decay of transcripts with premature stop codons, and the same accounts for the recruitment of the m6A ‘reader’ protein YTHDF2, leading to accelerating degradation of those transcripts [31,43].

Delving into the molecular mechanisms and pathways influenced by the m6A RNA modification may provide insights into potential therapeutic targets and novel prognostic biomarkers for hematologic malignancies. The m6A RNA modification has been reported as an important factor assisting the balance between self-renewal and differentiation of HSCs. Dysregulation of m6A-modifying enzymes, such as the m6A RNA mark’s ‘writers’ (m6A methyltransferases METTL3, METTL14, WTAP, RBM15, VIRMA, ZC3H13, etc.), ‘erasers’ (m6A demethylases FTO and ALKBH5) and ‘readers’ (m6A-binding proteins YTHDF1/2/3, YTHDC1/2, HNRNPs, IGF2BP, eIF3, etc.), influence HSCs’ gene expression programs. Dysregulated expression profiles of several modifying enzymes often lead to malignant phenotypes within the hematopoietic compartment [32,44,45,46]. The relevance of m6A modification in various studies on hematological malignancies underscores the importance of proper intracellular localization of m6A-modifying enzymes in maintaining normal hematopoietic homeostasis [24,47]. Aberrantly modified m6A RNA molecules contribute to immune evasion and resistance mechanisms, which is commonly observed in B-cell lymphomas, by altering the stability and translational output of key immune and oncogenic transcripts. Upon activating signals, the m6A demethylase ALKBH5 and lncRNA-treRNA1 (translation regulatory lnc-RNA1) translocate within the nucleus, where they form a functional complex with the RNA helicase DDX46. The ALKBH5/lnc-treRNA1/DDX46 nuclear complex facilitates the removal of m6A modifications from key B-cell receptor (BCR)-related transcripts, thereby providing fine-tuning to BCR expression levels. Disruption of this axis leads to impaired transcript processing and diminished BCR-related gene expression essential for B-cell functionality [48]. Furthermore, m6A demethylase ALKBH5, is frequently enhanced in human diffuse large B-cell lymphoma (DLBCL, the most common and aggressive form of non-Hodgkin lymphoma) samples directly demethylates BCR signature transcripts, hijacking the post-transcriptional/translational regulation of gene expression, which is crucial for normal B-cell development and lymphomagenesis [49] (Figure 1B).

The m6A demethylases FTO (fat-mass and obesity-associated protein) and ALKBH5 have distinct subcellular localizations and largely non-overlapping targets. Their activity is modulated by intermediary metabolism, affecting RNA demethylation processes. The dynamic nature of epitranscriptome is in part influenced by the subcellular movement of FTO, which shuttles between the cytosol and the nucleus, in contrast to the more stable nuclear localization of ALKBH5 [33,50]. Studies on AML and B-cell lymphomas have demonstrated that FTO, a multifunctional RNA demethylase, is capable of removing several types of methyl groups (including m6A) from diverse RNA classes. FTO localization patterns across the nucleus or cytoplasm enable selective RNA demethylation. Nuclear FTO preferentially targets m6A in pre-mRNAs and snRNAs, whereas cytoplasmic FTO accesses mainly cap-adjacent m6Aₘ on mature cytoplasmic mRNAs. As a result, FTO knockdown increases m6A, affecting RNA function in different cell compartments and further implying an emerging pattern for oncogenic transformation via subcellular mis-localization of FTO [33]. ALKBH5 activity, on the other hand, is regulated by post-translational modifications, such as ubiquitination. A recent study reports that USP36, a deubiquitinating enzyme, is associated with and promotes ALKBH5 stabilization, thus contributing to tumorigenesis [51]; in the same direction, ALKBH5 deficiency was suggested to represent an inhibiting factor for tumor proliferation in AML [52].

ALKBH5 and FTO are also (αKG) alpha-ketoglutarate-dependent enzymes, which are potentially inhibited by high accumulation of D-2-HG (D-2-hydroxyglutarate), a natural metabolite that is aberrantly produced by IDH1/2 (isocitrate dehydrogenases 1/2), proteins frequently found mutated in primary AML. D-2-HG competitively inhibits the activity of αKG-dependent dioxygenases, leading to a m6A hypermethylation profile in AML [53,54]. Along with αKG, other Krebs cycle metabolites can also alter the activity of m6A demethylases FTO/ALKBH5 [55,56]. FTO and ALKBH5 are shown to be enriched or depleted in the nucleus, when *O*-GlcNAcylation (bonding of a single *O*-linked N-acetylglucosamine moiety to a serine or threonine residue) is elevated or suppressed, respectively, suggesting that deposition of *O*-GlcNAc may also influence the epitranscriptome. *O*-GlcNAcylation is a ubiquitous post-translational protein modification influencing the subcellular compartmentalization of several proteins, and its elevation within the cell is regulated by *MYC* expression and αKG exposure. The exact protein network intervening in this mechanism merits further elucidation [55]. The transcription factor MYC binds to the *IDH2* promoter to transcriptionally activate it, leading to an increased αKG cellular pool. Along with *IDH2*, MYC also regulates various enzymes that positively influence anaplerotic mitochondrial metabolism, glycolysis, and the transport of glucose and glutamine through the cell membrane, suggesting its critical role in cancer-associated metabolic reprogramming and representing an important mechanism of nuclear–mitochondrial crosstalk [57]. Data from AML show that *IDH1/2* mutations lead to increased D-2-HG production, which competes with αKG, resulting in αKG depletion and inhibition of ALKBH2 and FTO activity (Figure 2A). Also, other αKG-dependent enzymes, such as DNA demethylases TET-1 and 3, under circumstances of decreased αKG, experience adverse effects, compromising their activity [53,54,55,58].

In acute lymphoblastic leukemia (ALL), either adult or pediatric type, as well as in multiple myeloma (MM), altered m6A modification patterns have been reported to affect malignant cell survival and proliferation [59,60,61,62]. In AML, the m6A ‘reader’ YTHDC1 has been found in elevated amounts in the nucleus and forms nuclear speckles and super enhancer condensates. m6A chemical marks on RNA molecules are a significant requirement for YTHDC1 to form nuclear YTHDC1–m6A condensates (nYACs), creating a nest for protecting m6A-mRNAs from the PAXT-complex (Poly(A) Tail eXosome Targeting complex) and the related exosome-associated RNA degradation. Protection and stabilization of m6A transcripts accelerates cell survival and proliferation [63] (Figure 2A).

Inhibitors targeting m6A RNA “writer” METTL3 and demethylases FTO and ALKBH5 are under constant investigation as they represent promising targets for fine-tuning of cells’ RNA epigenomes [64]. Inhibitors of METTL3/FTO/ALKBH5 related only to hematologic malignancies are presented in Table 1.

### 2.2. 5-Methylcytosine (m5C) RNA Modification

5-Methylcytosine (m5C) RNA modification refers to a post-transcriptional chemical addition of a methyl group to the fifth carbon of cytosine residues. m5C modification occurs in all (already known) kinds of RNA molecules (mRNAs, rRNAs, tRNAs and regulatory ncRNAs) and is distributed in non-random patterns. Its presence is detected in all endo-cellular compartments, including the nucleus, cytoplasm and mitochondria [25,26]. mRNA sequences are enriched with m5C modification within 5′ and 3′UTRs (untranslated regions), prevailing mainly near the translational start codon, primarily affecting protein synthesis rates and, as a consequence, overall gene expression profiles [73]. Enzymes involved in m5C RNA modification are categorized as ‘writers’, with distinguished members being the NSUN1-7 (NOP2/Sun RNA Methyltransferase) family and *TRDMT1* (TRNA Aspartic Acid Methyltransferase 1 or DNMT2), which preferentially methylate tRNA cytosines [74,75]. NSUN2 modifies cytoplasmic tRNAs, mediating their cleavage and stability, particularly under stress conditions. It also methylates non-coding RNAs, affecting their maturation process [76,77]. Another category comprises the m5C ‘erasers’, which remove methyl groups from RNA, reversing the m5C modification. TET (Tet Methylcytosine Dioxygenase) proteins belong to this family. They are primarily known for their role in DNA demethylation; however, they are also implicated in RNA demethylation [78]. Finally, ‘readers’ are proteins that recognize and bind to RNA-methylated cytosine residues, influencing downstream processes. Among them ALYREF recognizes and binds to m5C-modified RNAs, mediating their nuclear export. Depletion of this m5C ‘reader’ protein causes nuclear retention of m5C-methylated transcripts [34]. Additionally, YBX1 (Y-Box Binding Protein 1) ‘reader’ protein interacts with m5C-modified RNAs to regulate their translation and stability, influencing various aspects of RNA metabolism. This m5C-binding protein ensures the expansion of HSCs, by stabilizing m5C-modified mRNAs essential for cell proliferation. Disruption in YBX1 function can impair HSC maintenance and differentiation [27].

The relevance of m5C RNA modification in hematological malignancies is an area of active research, with emerging evidence highlighting its impact on the development and progression of these disorders. *TET2*-deficient HSCs manifest a global increase in chromatin accessibility and genome instability, finally favoring the development of myeloid malignancies [79,80]. Beyond its established function in DNA demethylation, TET2 in the nucleus oxidizes m5C in RNA, especially in retrotransposon-derived RNAs. This oxidation prevents the binding of MBD6 (methyl-CpG-binding domain protein) to chromatin-associated retrotransposon RNA (carrying m5C modifications), which further recruits deubiquitinases that remove mono-ubiquitination from histone H2A at lysine 119 (H2AK119ub), leading to an “*open chromatin*” conformation. Briefly, TET2 oxidizes m5C and antagonizes this MBD6-dependent H2AK119ub deubiquitination. Loss of TET2 activity leads to open chromatin conformation globally, promoting increased gene transcription in HSCs, which contributes to leukemic transformation, explaining the paradox of the highly demethylated (instead of methylated) DNA status in *TET2* deficient HSCs [80] (Figure 2B). Confirming the above is the observation that *TET2* mutations impacting its expression levels occur very frequently (about 15%) in patients with various myeloid neoplasias [81]. Expression levels of m5C regulators may serve as biomarkers for AML prognosis and treatment response, while *NSUN2*, *NSUN5*, and *TET2* inhibitors are suggested to act as potential modulators in an effort to restore normal HSCs’ methylation patterns by reversing the malignant phenotype [82].

Mitochondrial dysfunction is a hallmark of several hematological malignancies, and mitochondrial RNA modifications contribute to the regulation of proper mitochondrial function. A recent study focused on the mitochondrial RNA methyltransferase METTL17, which catalyzes m5C modifications on mitochondrial RNAs (mainly mt-ribosomal RNA), enhancing translation efficiency and supporting the assembly of oxidative phosphorylation (OXPHOS) complexes. Elevated METTL17 expression in AML cells leads to increased OXPHOS activity, shifting metabolism from glycolysis to oxidative phosphorylation, therefore satisfying the high energy demands of proliferating leukemic cells. METTL17 (Knock Out) KO cells have exhibited significant inhibition of AML cell proliferation, induced cell cycle arrest and concurrent reduced OXPHOS [83].

Although most functional studies implicate m5C RNA modification in tumorigenesis, design of antagonists or inhibitors of modifying enzymes responsible for m5C incorporation in RNA molecules and recognition has not yet yielded results in clinical hematology.

### 2.3. Adenosine to Inosine (A → I) RNA Modification

Adenosine deaminases acting on RNA (*ADARs*) catalyze adenosine to inosine (A → I) editing on double-stranded RNA (dsRNA). ADAR family includes the catalytically active enzymes ADAR1 and ADAR2 and also the non-enzymatic ADAR3 [84]. Two key isoforms of *ADAR1* have been recognized; ADAR1p110, which is nuclear, and ADAR1p150, which is driven by a type I interferon (*IFN*) inducible promoter. ADAR1p150 is mainly present in the cell cytoplasm, enabling regulation of cytoplasmic dsRNA and immune-sensing pathways [85,86]. Alternatively, the ADAR1p110 isoform is regulated by a constitutive promoter, and both ADAR1p110 and ADAR2 function on dsRNA substrates within the nucleolus [84]. Also, ADAR1p110 facilitates resolution of telomeric R-loops required for cancer cell proliferation, implying proto-oncogenic properties [87]. ADAR2 has been reported to be significantly downregulated in hematologic patients with translocation [t(8; 21)] or inversion [inv(16)] mutations, further suggesting a molecular mechanism whereby hypo-editing of specific transcripts by ADAR2 is implicated in the pathogenesis of t(8; 21) AMLs [88].

A → I biochemical transformation of RNA adenosines acts as a functional A → G mutation, influencing splicing, microRNA targeting, coding capacity, and RNA structure [89]. In hematologic malignancies, the clearest localization-linked disease signal is enrichment of the cytoplasmic ADAR1 isoform p150 pool, which excessively edits cytoplasmic dsRNA and suppresses innate dsRNA sensors and immune pathways. A → I editing within inverted Alu elements and other dsRNA regions masks them from innate sensors like melanoma differentiation-associated protein 5 (*MDA5*), dampening type I interferon responses and preventing immune activation, a mechanism often hijacked in a wide variety of cancers [90,91]. In hematologic neoplasias, this promotes leukemic stemness, disease relapse, and immune evasion in leukemia, lymphoma, and multiple myeloma (MM) [92].

Global hyper A → I editing or elevated ADAR1 expression often correlates with worse prognosis in AML. Levels of RNA A → I editing observed among AML genotypes vary, with patients carrying *CEBPA* and *NPM1* mutations having lower levels in contrast to patients carrying mutated chromatin regulators and *TP53*. Notably and applying to all cases, increased levels of RNA A → I editing are a predictor of poor survival and lack correlation with elevated ADAR1 expression [93]. Expression of ADAR1 (both isoforms) and ADAR2 enzymes in human myeloid leukemia cells during differentiation revealed that ADAR1 is progressively modulated during maturation and ADAR2 undergoes a sharp increase during differentiation [94].

In acute lymphoblastic leukemia (ALL), especially T-ALL, elevated A → I editing has been associated with relapsed T-ALL; ADAR1 editing by p150 isoforms (cytoplasmic) attenuates dsRNA sensing and promotes leukemic survival through both MDA5-dependent and independent pathways based on the intrinsic expression of MDA5 [95]. The ADAR1p150 isoform has been suggested as the main regulator of the MDA5 pathway and is the major contributor to leukemia-initiating cells (LIC) generation in myeloid leukemia, constituting a small, self-renewing subpopulation of leukemic cells responsible for disease initiation, maintenance, and relapse. In addition, the Zα-RNA-binding region specific to the ADAR1 p150 isoform is responsible for the induction of IFN-stimulated genes (ISGs) in HSCs [91,96,97]. An increased ADAR1 p110/p150 ratio is also characteristic in DLBCL of the most common and aggressive types of B-cell lymphomas and is associated with late disease stages [98].

In CLL cohorts, studies have mapped recoding A → I events and miRNA/3′UTR editing, with some edited sites exhibiting a prognostic potential [99]. Moreover, in multiple myeloma, copy number amplification of chromosome 1q21, which contains both *ADAR1* and interleukin-6 receptor (*IL-6R*) gene loci, is associated with a poor prognosis, while *ADAR1* knockdown alters sensitivity to lenalidomide [100]. A → I RNA editing has been implicated in various biological processes, including immune and stress responses, HSCs’ fate determination, hematopoietic cell development, and hematologic malignancy progression [101]; however, deeper insights into the subcellular localization of these events require further elucidation. Translation to patient therapies remains at an early preclinical stage due to some limitations that have been only recently altered with the advent of deep sequencing technologies.

### 2.4. Pseudouridine (Ψ) RNA Modification

Pseudouridine (Ψ) is one of the most abundant RNA modifications, displaying chemical properties distinct from the naturally occurring nucleotides. It is found in various types of RNA, including transfer (tRNA), ribosomal (rRNA), small nuclear (snRNA) and messenger (mRNA). The modification involves the isomerization of uridine to pseudouridine catalyzed by either RNA-independent or RNA-dependent pseudouridylation [102]. RNA-independent pseudouridylation is performed by stand-alone pseudouridine synthases (*PUSs*) based on structural or sequence recognition motifs of their target RNA substrates. On the other hand, RNA-dependent pseudouridylation relies on a complex involving a box H/ACA RNA, along with four core proteins, the Ψ synthase dyskerin (*DKC1*), glycine-arginine-rich protein 1 (*GAR1*), non-histone protein 2 (*NHP2*), and nucleolar protein 10 (*NOP10*), which form a box H/ACA small ribonucleoprotein (RNP) [28,103]. PUSs are widespread within all cell compartments, including the nucleus, cytoplasm, and mitochondria, and their relocalization upon stimuli can influence the RNA substrate pool available for pseudouridylation [104]. Based on its highly conserved nature and abundance within cell, Ψ is believed to be functionally very important, since it enhances the stability of RNA molecules by improving their base-pairing capabilities and structural rigidity.

This modification has very early been recognized as a critical feature for tRNA folding and is located in nearly all tRNAs within the TΨC stem loop, while other, less frequent Ψ sites are also found at different tRNA positions [105,106]. At the three-dimensional level of the ribosomal subunit interactive surface, Ψ is highly concentrated in functionally important sites [107,108]. The same accounts for Ψ locations within spliceosomal snRNAs U1, U2, and U5, all of which participate in spliceosome assembly and pre-mRNA splicing reactions [109]. SnRNA U2 is extensively modified, featuring 14 Ψs in conserved residues, which contribute to snRNP and spliceosome assembly during the splicing process. SHQ1, a critical factor for H/ACA snoRNP assembly, is overexpressed in T-cell acute lymphoblastic leukemia (T-ALL) and directly activated via the oncogenic NOTCH1. Elevated SHQ1 facilitates U2 snRNA pseudouridylation and efficient global pre-mRNA splicing, hence supporting T-cell leukemogenesis [110]. Pseudouridylation of snRNAs (U2, U4, U5) ensures correct spliceosome function. Loss of DKC1 or snoRNA-guided pseudouridylation impairs splicing fidelity, possibly leading to aberrant splicing of tumor suppressors or oncogenes, exon skipping, or intron retention, all known as crucial features in hematological cancers, especially in MDS with splicing factor mutations (e.g., SF3B1, SRSF2).

Artificially enriched with Ψ mRNA, encoding erythropoietin has been shown to increase its translation efficiency [111]. Pseudouridylation has also been detected in several miRNAs, including members of the let-7 family, evolutionary conserved microRNAs that mediate post-transcriptional gene silencing, aiming to regulate/attenuate various normal biological processes, but also induce tumor suppression. Ψ is embedded in pre-miRNAs by PUS1 and/or TruB1, promoting their maturation [112,113]. In addition, pseudouridylation of various mitochondrial tRNAs (mt-tRNACys, mt-tRNASer and mt-tRNATyr) induced by the P175fs mutation of *PUS1* reduces mitochondrial protein synthesis, contributing to mitochondrial dysfunction and inducing mitochondrial myopathy, lactic acidosis, and sideroblastic anemia syndrome (MLASA), manifesting as a severe defect in erythropoiesis [114].

Pseudouridine’s contribution to hematological malignancies has been documented by several studies. HSCs from patients with myelodysplastic syndrome (MDS) are highly concentrated with a specific subset of tRNA-derived fragments (tRFs), termed mini tRFs, with a characteristic 5′ terminal oligoguanine tail (mTOGs), which frequently undergoes pseudouridylation (Ψ), enhancing their stability and function. Pseudouridylated mTOGs interact with the RNA-binding protein PABPC1. This interaction disrupts the recruitment of the translational co-activator PAIP1 (Poly(A)-binding protein-interacting protein 1), leading to selective repression of mRNAs with pyrimidine-enriched sequences (PES) in their 5′ untranslated regions (UTRs), including 5′ terminal oligopyrimidine tracts (TOP). These mRNAs typically encode components of the protein synthesis machinery. In high-risk MDS patients, dysregulation of mTOGs results in increased translation of pyrimidine-enriched sequences (PES) containing mRNAs, contributing to aberrant protein synthesis and impaired differentiation of HSCs. This dysregulation is clinically associated with progression to acute myeloid leukemia (AML) and reduced patient survival [115].

Abnormal hematopoiesis associated with *DKC1* mutations may also arise from aberrant rRNA pseudouridylation. Profiled snoRNAs from patients with X-linked dyskeratosis congenita (X-DC), a multisystem disorder characterized by multiple *DKC1* mutations and bone marrow failure, uncovered highly heterogeneous snoRNA landscapes in CD34+ cells. Ψ levels at specific uridine residues were correlated with snoRNA levels that guide pseudouridylation at those sites. Reduced rRNA pseudouridylation can lead to defective ribosome biogenesis and impaired translation of mRNAs, especially IRES-containing transcripts (e.g., p53, c-MYC), promoting skewed protein synthesis. Wild-type *DKC1* overexpression could largely rescue normal snoRNA levels and restore faithful CD34+ progenitor differentiation into mature blood lineage-specific cells [116].

Conclusively, defective pseudouridylation disrupts RNA metabolism at multiple levels, such as translation, splicing, ribosome function, and all these perturbations can initiate or aggravate the course of several hematological malignancies.

### 2.5. 2′-O-Methylation RNA Modification

This RNA modification involves the methylation of the 2′-O position of the ribose sugar in RNA, catalyzed by the fibrillarin and NOP56 complex. It influences RNA stability, splicing, and protein synthesis. Nucleophosmin 1 (*NPM1*) participates in rRNA 2′-O-methylation by direct binding to several C/D box small nucleolar RNAs and to the methyltransferase fibrillarin. The fibrillarin–NPM1–small nucleolar RNA complex actively methylates rRNA. NPM1 loss of function, a very frequent event in AML, results in altered rRNA 2′-O-methylation and dysregulated translation [35].

2′-O-methylation of rRNAs, guided by snoRNAs and executed by fibrillarin (FBL), is dynamically remodeled in AML, and specific methylation patterns on the ribosome surface correlate with leukemia stem cell (LSC) signatures. Enhanced 2′-O-methylation is reported to rewire translation towards amino acid transporter mRNAs enriched in optimal codons, increasing intracellular amino acid levels and supporting the LSC phenotype [36].

## 3. Consequences from Altered Subcellular Topology of Non-Coding (nc) RNA Molecules

The regulatory potential of many ncRNAs has been revealed at specific subcellular locations. Nevertheless, many ncRNAs are discretely distributed in different cellular compartments, and the related biological significance remains largely unclear.

### 3.1. Intercellular Distribution and Function of Micro-RNAs (miRs)

Among the well-documented functions of mature miRs (21–23 nt) in the cytoplasm is the targeting of mRNA molecules, mainly at the 3′ UTR, leading to their degradation. To accomplish this aggressive function against mRNAs, miRs are tethered with catalytic Argonaute (Ago) proteins and are assembled into RISC (RNA-induced silencing) complex [117]. Pre-miRNAs with a hairpin 2D construction are transported to the cytoplasm through their nuclear transport receptor exportin-5 and another karyopherin, CRM1, which mediates their nuclear–cytoplasmic shuttling [118]. Importin 8, a member of the karyopherin β transporters on the nuclear envelope, has been reported to participate and facilitate miRs’ relocation within cell nuclei [119]. Despite their clearly defined role in the cytoplasm, numerous mature miRs have also been reported to selectively relocate within cell nuclei, having been assigned with gene expression regulatory roles. These miRs encompass variations of a 6-nucleotide motif [120] responsible, but not sufficient, for their reimport in the nucleus. These motifs have been evident in the 5p-arm of let-7 family members [121].

Studies that differentiate the intracellular localization of miRs between the nucleus and the cytoplasm are relatively few. A basic reason for that is the important methodological difficulties arising from RNA extraction procedures from different cell fractions: the nucleus, cytoplasm and mitochondria. Pure RNA extraction preventing cross-contamination is challenging, mostly due to insufficient gradient methods that result in incomplete separation and consequent overlap between organelle fractions. Also, RNases can act during processing, especially if cell compartments are not promptly stabilized, leading to massive RNA degradation. Nevertheless, a well-defined function of nuclear miRs is the regulation of other miRs’ biogenesis [122]. Some studies have expanded miRs’ silencing mechanism in the nucleus and suggest that this is assisted by GW182 family proteins (components of miR-induced silencing complexes) and by Ago proteins that translocate in nuclear foci to target miRs and nuclear long non-coding RNAs (lncRNAs), such as MALAT-1 [123]. Analysis of subcellular nuclear and cytoplasmic compartments of neural stem cells derived from the human embryonic cell line WA09 revealed miRs with consistently stronger standard scores in the nuclear fraction, namely hsa-miR-30b/c, hsa-miR-374a/b, and hsa-miR-19a/b, as well as different miR species significantly enriched in the cytoplasmic fraction, such as hsa-miR-20a/b, hsa-miR-17, hsa-miR-106a/b, hsa-miR-93, etc. [124].

In the hematopoietic compartment, hsa-miR-223 represents a well-documented double functioning molecule. When this miR is localized in the nucleus, it silences the *NF1A* (Nuclear Factor 1A) promoter via recruitment of the polycomb repressive complex (PcG), which is responsible for the trimethylation of H3K27. In the cytoplasm of myeloid progenitors, hsa-miR-223 targets the *NF1A*-mRNA for degradation during granulopoiesis [14,15]. Conclusively, hsa-miR-233 is devoted to silencing the *NF1A* factor, with a dual role in both the nucleus and the cytoplasm of myeloid progenitors. Also, hsa-miR-223 and C/EBPa (CCAAT Enhancer Binding Protein alpha) regulate the expression of *E2F1* in AML. E2F1 upregulates hsa-miR-223 transcription via promoter binding, and cytoplasmic hsa-miR-223 suppresses *E2F1*-mRNA. In AML cells, this feedback loop is disrupted, resulting in aberrant expression levels of E2F1 and hsa-miR-223, contributing to uncontrolled cell proliferation [125]. Several other miRs, hsa-miR-690, hsa-miR-706, and hsa-miR-709, have been reported to be localized within the nuclear compartment, regulating or fine-tuning other miR species’ expression patterns [16]. Hsa-miR-690 is enriched in the nucleus of myeloid cells and, upon its overexpression, attenuates the expression of the C/EBPα protein, which is involved in the development of granulocyte–monocyte progenitors, affecting their subsequent terminal differentiation [126]. Loss of C/EBPα leads to increased myeloid proliferation [127].

Loss or downregulation of HOX regulatory miRs, such as hsa-miR-204 and hsa-miR-128a, and overexpression of Homeobox (*HOX*) genes (specifically HOXA and HOXB) have been suggested as potential disease-specific signatures in AML patients simultaneously carrying mutations in the nucleophosmin 1 (*NPM1*) gene. At steady-state, NPM1 usually resides in the nucleolus, whereas only a minimal fraction shuttles between the other cell compartments [11]. *NPM1* mutations in AML result in the trans-localization of NPM1 protein into the cytoplasm, representing the most common (50–60%) genetic alteration in AML-NK (Normal Karyotype) and in about one-third of all AML cases [12,13]. Quantification of the precise proportion of NPM1 protein localized to the cytoplasm of *NPM1*-mutated cells is challenging, and the biological relevance of how *NPM1* mutant cells are prone to develop acute leukemia by acting at the chromatin level in cooperation with miR species (hsa-miR-204 and hsa-miR-128a) as potential partners remains unclear. This active network can include lncRNAs, which is discussed in the lncRNA section.

Studies in MDS and AML have shown substantial discrepancies in miR expression profiles, and a potential etiologic factor for these discrepancies may be the different cell sources used for RNA extraction. Unsorted total BM cells, BM mononuclear cells, and selected CD34+ BM cells have been used as a primary source for RNA isolation and miR detection, with each investigative approach having both advantages and disadvantages. MiRs secreted into the extracellular fluids and imported into exosomes are also subject to investigations, mainly for biomarker identification. Analyses of the unsorted BM cells can deliver more information about the microenvironment and impaired erythroid differentiation than about the more homogeneously isolated CD34+ BM cell population, which captures in more detail the clonal, MDS-specific blast cell population and demonstrates the profound, intrinsic changes in miR expression profiles. However, significant differences in miR expression have been reported between healthy controls and patients with MDS and AML, independent of the cell source [128]. The most significantly down- or upregulated miRs found in patients with MDS and AML are listed in Table 2. These observations support the idea of existing absolute specializations in the creation of molecular networks/interactions within functionally different cells, even in the same tissue.

Mutations in genes encoding spliceosome components occur in more than 50% of MDS patients, with *SF3B1* (Splicing Factor 3b Subunit 1)*, SRSF2* (Serine and Arginine Rich Splicing Factor 2), *U2AF1* (U2 Small Nuclear RNA Auxiliary Factor 1), and *ZRSR2* (Zinc Finger CCCH-Type, RNA Binding Motif and Serine/Arginine Rich 2) as the most frequently mutated [129,130]. Aberrant splicing as a consequence of spliceosome mutations have been shown to affect mRNA transcripts and miR expression mainly during their post-transcriptional maturation. In particular, pri-miRNAs encoded within introns of coding genes, termed as miRtrons, are processed by the nuclear splicing machinery like typical introns to form intra-nuclear stable hairpins with a shorter stem feature before they are exported to the cytoplasm [131]. Downregulation of tumor suppressive miRs such as the let-7 family, miR-423, and miR-103a has been documented in MDS patients carrying spliceosome mutations [132], which can be associated with impaired splicing events in the nucleus.

Among mitochondrial ncRNAs, mitochondrial microRNAs (mt-miRs) are transcribed directly from the mitochondrial genome as their nuclear counterparts and regulate mitochondrial gene expression post-transcriptionally [133]. Along with mt-miRs, nuclear-derived miR species are imported into the mitochondria to modulate the expression levels of mRNAs originating from the mitochondrial genome. One of the key proteins implicated in this complex transport process is Argonaute2 (Ago2) [134]; however, mitochondrial miR-derived species and hematologic malignancies still remain unconnected.

Another category of small nuclear ncRNAs, Piwi-interacting RNAs, have been framed as regulatory molecules in hematology. In DLBCL, upregulation of the piRNA-30473 has been shown to induce increased expression of methylation factor *WTAP* (Wilms’ tumor 1-associating protein) in the nucleus and increased m6A levels mainly in (hexokinase2) *HK2* mRNA, contributing to high glycolytic rates and promoting tumorigenesis [135].

### 3.2. Abundance of Long Non-Coding RNAs (lncRNAs) in the Subcellular Compartments

Long non-coding RNAs (lncRNAs) are long transcripts (mainly ≥ 200 nt) that do not code for proteins, but instead play roles in regulating gene expression, modulating chromatin dynamics by recruiting histone-modifying and chromatin-remodeling complexes, and RNA splicing. Recent evidence reveals that some lncRNAs actually harbor small open reading frames (sORFs) that are actively translated into functional micropeptides, but their actual relevance in promoting malignant phenotypes remains uncovered [136]. LncRNAs associated with chromatin remodeling are highlighted as important regulators of normal and malignant hematopoiesis [137]. Beyond their chromatin remodeling properties, concerns have arisen about lncRNAs’ transportation to different locations within a cell and about the consequences of this in cell homeostasis and performance.

Distribution of lncRNAs among the different endo-cellular clusters has been determined by cell fractionation, followed by RNA-seq, and has been shown to be proportionally higher in cytoplasmic ribosome-enriched clusters than in nuclear clusters. TUG1-lncRNA, which is involved in the upregulation of the *HOX* family and other growth-control genes in the nucleus, has also been detected in cytosolic cell fractions containing five or six ribosomes. Furthermore, DANCR and H19 lncRNAs have shown clear enrichment in the cytoplasm over the nucleus, providing evidence for other, still unrecognized functions within the cell, and more specifically, implication of lncRNA in the regulation of the translation process [138]. However, there is a debate concerning the precise lncRNA endo-cellular compartmentalization, since other researchers propose that they preferentially localize and function within the cell nucleus [139].

The cytoplasmic translocation of the *NPM1* mutants in AML patients is accompanied by upregulation of HOXA/B genes and alters the hematopoietic cells’ genome topology, as previously mentioned [12]. In this complex microenvironment, specific lncRNAs deriving from the *HOXA/B* loci (on chromosomes 7 and 17, respectively), are also implicated. The anterior *HOXB*-derived lncRNA-HOXBLINC (Gene ID: 128462380) is activated in AML and promotes HSC transcription signatures by exploiting an active 3D interactome to recruit the MLL1/Setd1a complex and deposit H3K4me3 marks on each of the anterior *HOXB* genes for their activation [140]. LncRNA-HOTAIRM1 is hosted within the *HOXA* genomic cluster, between the *HOXA1* and *HOXA2* genes. Its significantly higher expression in AML patients carrying *NPM1* mutations attributes to lncRNA-HOTAIRM1 a great prognostic impact [17]. LncRNA-HOTTIP, which is transcribed from the distal *HOXA* locus, recruits the mixed lineage leukemia *MLL* family of histone methyltransferases (MLL1/WDR5 complex) to activate targeted genes in cis [141]. Given that both lncRNA-HOTTIP and NPM1 interact with CTCF (CCCTC-binding factor), a critical factor responsible for chromatin organization and for anchoring CTCF-flanked topologically associated domains (TADs) [18,19], this makes it particularly interesting and important to determine deeper mechanisms of action for NPM1 mutants and mis-localization of NPM1 associations within the evolutionary process of leukemogenesis (Figure 3A).

In AML with mutated *NPM1*, more evidence exists for the implication of lncRNA translocations. lncRNA-LONA shifts from the cytoplasm to the nucleus, alongside the mutant NPM1 export, which moves towards the opposite direction. Within the nucleus, lncRNA-LONA alters myeloid differentiation by regulating tissue-specific mRNAs. This mechanism promotes leukemogenesis and reduces chemotherapy response in vivo [142]. The involvement of lncRNAs in m6A RNA modification pathways has also been identified in AML. LncRNA-UCA1 has been reported to contribute to AML progression by upregulating the expression of *CXCR4* (C-X-C Motif Chemokine Receptor 4) and *CYP1B1* (Cytochrome P450 Family 1 Subfamily B Member 1), through binding and stabilization of the m6A methyltransferase METTL14. This interaction affects m6A levels and influences localization and function of METTL14 in AML cells [143]. Another study has demonstrated that the lncRNA-MALAT-1 binds to METTL14, promoting m6A modification of *ZEB1*-mRNA (Zinc Finger E-Box Binding Homeobox 1). This interaction enhances the aggressiveness of AML by influencing the localization and function of METTL14 intranuclearly [144] (Figure 3B). Within the same context, the nuclear METTL16 has been reported to interact with the 3′ triple-helix structure of MALAT-1, both in vitro and in vivo. METTL16 is capable of catalyzing m6A modification on the lncRNA-MALAT-1 triple helix, which in turn enhances the accessibility of a U5-tract for binding by Heterogeneous Nuclear Ribonucleoprotein (hnRNP) C, a key player in pre-mRNA processing and maturation in the nucleus [145].

The translocation of non-coding RNAs (ncRNAs) between the nucleus and mitochondria is essential for regulating mitochondrial function and mitochondrial gene expression. Among lnc-RNAs identified in both nuclear and mitochondrial territories, nuclear lnc-RMRP has been identified as a component of the nuclear RNase MRP complex, mediating the processing of the short mature 5.8S rRNA [146]. In addition, lnc-RMRP in the cell nucleus interacts with telomerase to form a complex with RNA-dependent RNA polymerase, capable of synthesizing dsRNA precursors, which are processed by DICER1 into siRNAs [147]. Two RNA-binding proteins, HuR (human antigen R) and GRSF1 (G-rich RNA sequence-binding factor 1), are responsible for the mobilization of nuclear-encoded lncRNA-RMRP into the mitochondrial matrix. As a component of the mitochondrial RNase MRP, lnc-RMRP is important for mitochondrial DNA replication and RNA processing [148].

Another nucleus-derived lncRNA, linc-p21, has been shown to increase HIF-1a (hypoxia-inducible factor-1) stability with high specificity and consequently to promote glycolysis, affecting also mitochondrial OXPHOS [149]. Inhibition of the lncRNA-HOTAIR, a Hox Transcript Antisense Intergenic RNA encoded from the nuclear genome and imported into mitochondria, has also been reported to be associated with a remarkable decrease in UQCRQ (Ubiquinol-Cytochrome C Reductase, Complex III Subunit VII), inducing deficiency of complex III and impairment of the mitochondrial respiratory chain [150]. These studies highlight an as yet undiscovered field of interplay between endonuclearly expressed lnc-RNAs and mitochondrial regulation. In hematological malignancies, dysregulation of this transport system can contribute to altered cellular metabolism, apoptosis, and abnormal proliferation, affecting disease progression and treatment response, hence requiring deeper investigation.

LncRNAs are also transcribed from mitochondrial DNA and, additionally, some nuclear lncRNAs are transported into the mitochondrial matrix and influence mitochondrial gene expression and function. Mitochondrial long non-coding RNAs (mt-lncRNAs), transcribed from the mitochondrial genome, have emerged as key players in various cellular processes and have been implicated in the pathogenesis of several cancers. Their function within the mitochondrial compartment is not restricted to only metabolism reprogramming, but also extends to mitochondria-associated apoptosis, mitochondrial genome expression regulation, and stress signal transmission [151,152,153,154]. A circular non-coding RNA, mc-COX2, is produced from the *COX2* locus on the L-strand of the mitochondrial DNA and has been found at high levels in the plasma exosomes of patients with chronic lymphocytic leukemia (CLL), affecting disease recurrence and progression [155]. Another mitochondrial lncRNA, SncmtRNA, derives from mitochondrial 16S rRNA sequences and is consistently overexpressed in proliferating cells—both normal and malignant—but is absent in non-dividing cells, indicating its role in cell division. Two antisense lncRNAs, ASncmtRNA-1 and ASncmtRNA-2, with similar hairpin structures to SncmtRNA, are expressed only in normal proliferating cells and are downregulated in tumor cells, suggesting a potential tumor-suppressor function. Subcellular localization studies, using in situ hybridization, confocal, and electron microscopy, have revealed that SncmtRNA localizes predominantly to the nucleus and, more precisely, in the nucleoli and heterochromatin, in both normal and tumor cells. However, in tumor tissues, SncmtRNA shows variable localization, being nuclear in some cases and cytoplasmic in others, possibly reflecting its involvement in epigenetic regulation during malignant transformation. In contrast, ASncmtRNAs are localized to the nucleus in normal cells, but are notably absent in tumor tissues [153,156]. Conclusively, downregulation of SncmtRNA and the upregulation of ASncmtRNA-2 suggest a pivotal role of these transcripts in cell replicative senescence. ASncmtRNA-2 is linked to the production of the mitochondrial encoded miRs hsa-miR-4485-3p and hsa-miR-1973, with miR-4485 showing a significant increase. Overexpression of these mtmiRs in cells results in delayed progression through the G1 and G2/M phases of the cell cycle. These findings highlight a functional role for mitochondrial lncRNA ASncmtRNA-2 in generating regulatory miRs that influence cell cycle control [154].

## 4. Conclusions and Future Perspectives

There is clear evidence based on investigations, although still very limited, that non-coding RNA molecules localized in different membrane-separated cell compartments are prone to cooperating with different macromolecule networks, hence exhibiting alternate functions. “Genetic” over “epigenetic” events are considered the first and most important drivers for several hematological malignancies. However, epigenetic alterations are documented to occur early and sometimes initiate pathogenetic roles, especially in age-related clonal hematopoiesis and pre-leukemic states. The debate between these two pathogenetic mechanisms as driving factors for tumorigenesis is still under consideration and under continuous investigation. Nevertheless, “genetic” and “epigenetic” lesions are strictly interconnected; mutations in genes encoding modification enzymes that control the epigenome blur the line between “genetic” and “epigenetic” events [157]. Current technologies are increasingly capable of unraveling the spatial organization of RNA modifications and ncRNAs within cellular compartments, and these approaches are proving highly useful for understanding how alterations in RNA biology may contribute to tumorigenesis.

This review highlights defective endo-cellular pathways, referring to the aberrant topology of altered RNA modifications, mainly m6A, A → I, m5C, and Ψ, presenting an overview of factors triggering this phenomenon, and also the downstream pathways that are affected and stimulate the development of hematological malignancies. Mis-localized modified RNAs have been shown to be conditionally produced by relevant malfunctional modifying enzymes, affecting RNA maturation, stabilization, and interference to chromatin conformation (Figure 1 and Figure 2). These insights point to RNAs and their modifying enzymes as potential therapeutic targets in various hematological malignancies. Efforts to test and utilize new drugs targeting RNA-modifying enzymes are ongoing, since they are very promising therapeutic options over current therapies [158,159,160].

Beyond RNA modifications implicated in hematology, miRs and lncRNAs, the most recognized and studied ncRNA molecules, have emerged as critical epigenetic factors capable of disrupting chromatin structure and gene expression levels by misregulating either pre- or post-transcriptional tumor suppressors or oncogenes and altering RNA/protein interactions critical for cell integrity (Figure 3). With advances in delivery systems, RNA biology, and epigenetic understanding, ncRNA-based therapies are likely to become a part of future treatment options for several hematological malignancies. Particularly promising is their potential to effectively target malignant stem cells, overcome drug resistance, and act synergistically with existing therapies, targeting other dysregulated cellular pathways. A few attempts have entered clinical trials, such as the MRG-106 (Cobomarsen), an inhibitor of hsa-miR-155 tested in cutaneous T-cell lymphoma (CTCL) and other lymphomas; however, development was paused due to company restructuring [161]. Currently, there are some RNA-based therapies in clinical trials representing a growing and highly promising class of novel agents but they have not yet received approval from regulatory organizations as treatments for hematological malignancies. Treatments currently being tested in clinical trials specifically for blood cancers include antisense oligonucleotides (ASOs), small interfering RNAs (siRNAs), RNA aptamers, etc. As single-cell RNA-seq, epitranscriptomics, and AI-driven drug design evolves, new possibilities arise for the prediction of RNA structure–function interactions and tumor-specific RNA profiles, which will enable a “smart” RNA drug design, addressing many unmet needs for the effective treatment of hematological malignancies.

## Figures and Tables

**Figure 1 cimb-47-00758-f001:**
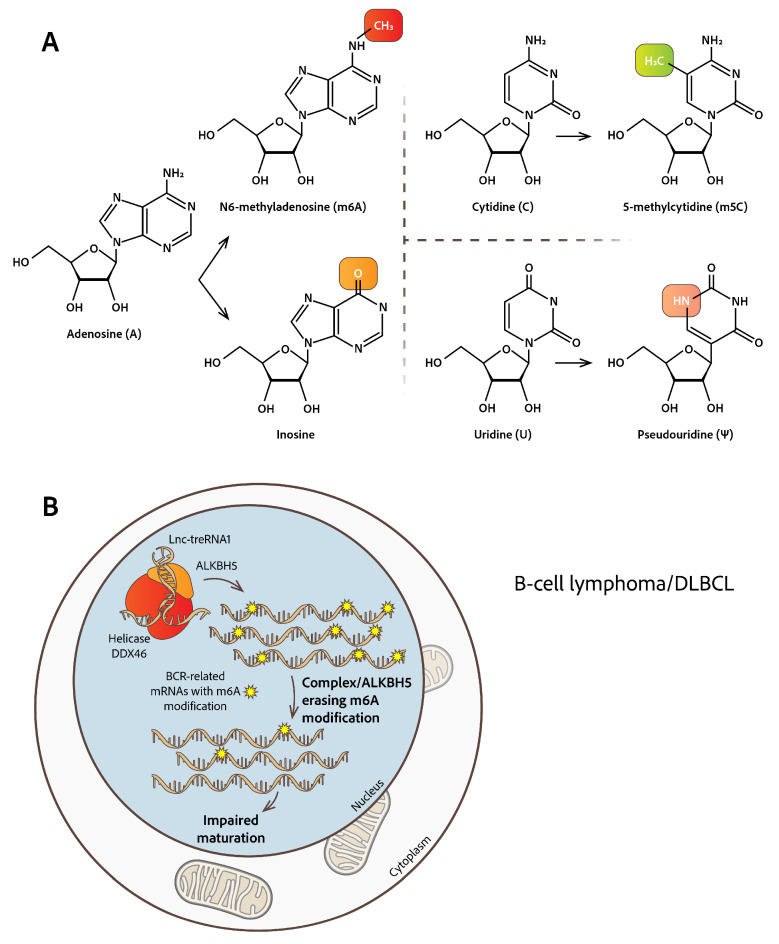
**RNA base modifications and m6A modification role in B-cell lymphoma.** (**A**) Chemical structures of selected RNA modifications including N6-methyladenosine (m6A), 5-methylcytidine (m5C), inosine (I), and pseudouridine (Ψ). (**B**) In B-cell lymphoma, m6A demethylase ALKBH5, lncRNA-treRNA1 and the RNA helicase DDX46 form a functional complex in the cell nucleus, affecting the removal of m6A modifications from key BCR-related transcripts with severe consequences to B-cell functionality. Also, ALKBH5 is frequently enhanced in human DLBCL and demethylates BCR signature transcripts, guiding lymphomagenesis.

**Figure 2 cimb-47-00758-f002:**
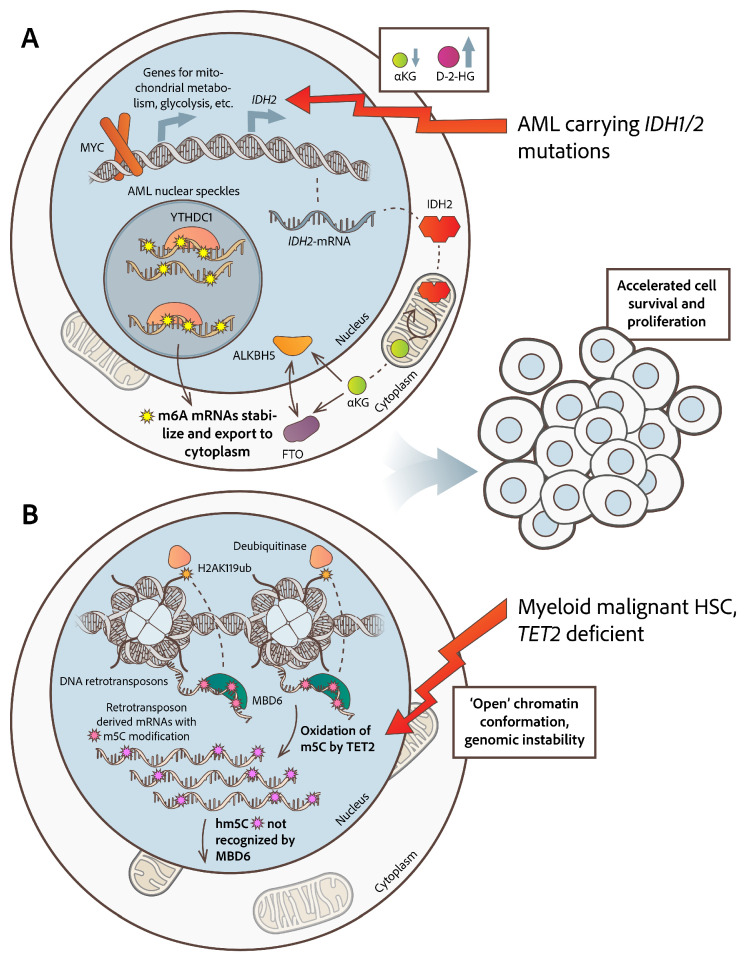
**Altered RNA modification pathways in myeloid malignancies**. (**A**) In AML, ALKBH5 and FTO are αKG-dependent enzymes and are inhibited by the high accumulation of D-2-HG, an alternative metabolite produced by *IDH1/2* mutants, leading to a m6A hypermethylation profile. MYC transcriptionally activates the *IDH2* promoter along with various other enzymes’ promoters which positively influence anaplerotic mitochondrial metabolism, glycolysis, and glucose transport, leading to an increased αKG cellular pool, suggesting its critical role in nuclear–mitochondrial crosstalk. Increased levels of the m6A ‘reader’ YTHDC1 in AML leads to the formation of nuclear speckles, protecting m6A-mRNAs from degradation. Protection and stabilization of m6A transcripts accelerates cells’ survival and proliferation. (**B**) In AML with mutated TET2, oxidation of m5C, especially in RNAs derived from retrotransposons, fails. Remaining m5C marks in RNA recruit MBD6 and deubiquitinases which remove mono-ubiquitination from H2AK119ub, leading to open chromatin status, increased transcription of retrotransposons, and genomic instability, all hallmarks of cells’ cancer state.

**Figure 3 cimb-47-00758-f003:**
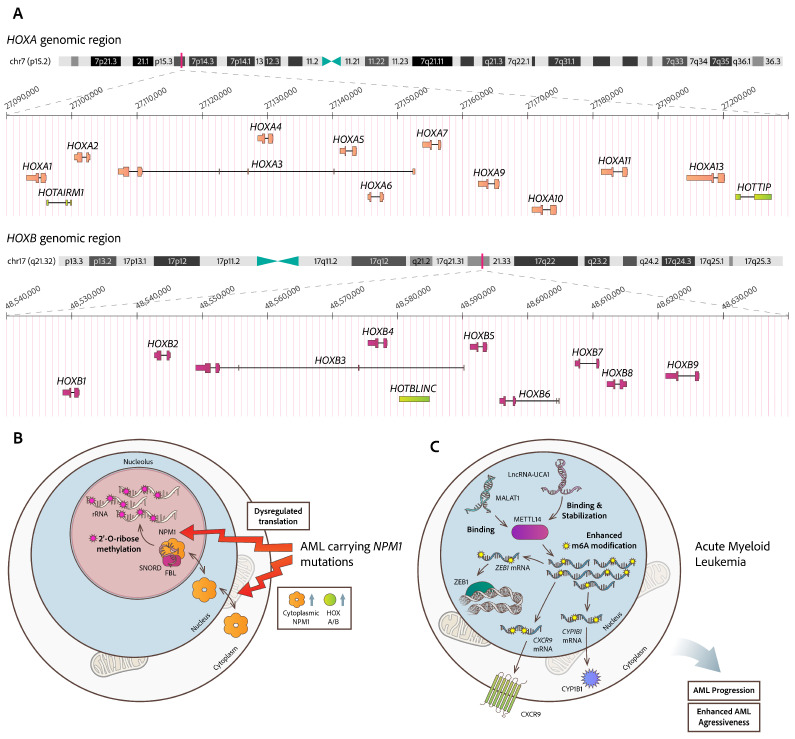
**Implicated lncRNAs in malignancies of myeloid origin**. (**A**) Genetic loci encompassing *HOXA/B* genes and expressed antisense lncRNAs. In AML patients carrying mutated NPM1, there is upregulation of the HOXA/B loci (on chromosomes 7 and 17, respectively) and of lncRNA-HOTAIRM1, lncRNA-HOTTIP and lncRNA-HOXBLINC (highlighted with light green), which are encoded from the same loci, but are reversely transcribed. Altered lncRNA expression profiles are implicated in several dysregulated pathways. (**B**) NPM1-mutated AML produces specific leukemia stem cell signatures by remodeling 2′-O-methylation of rRNAs, guided by snoRNAs and executed by fibrillarin (FBL). (**C**) LncRNA-UCA1 contributes to AML progression by upregulating the expression of CXCR4 and CYP1B1 and by promoting stabilization of the m6A methyltransferase METTL14. This interaction affects m6A levels and influences localization and function of METTL14 in AML cells. Also, lncRNA-MALAT-1 binds to METTL14, promoting m6A modification of ZEB1-mRNA and influencing the localization of METTL14 intranuclearly.

**Table 1 cimb-47-00758-t001:** m6A RNA modifying enzymes’ inhibitors as potential therapeutic options for hematologic malignancies.

Inhibitor	Target	Disease	Ref
STC-15 (Storm Therapeutics LTD), METTL3 inhibitor	Activates IFN signaling and remodeling of the TME towards pro-inflammatory state (ASCO Meeting Abstract, 2024).	Advanced malignancies	Clinical trial (NCT05584111)
Small molecule STM2457, METTL3/METTL14 inhibitor	Binds directly to the METTL3/METTL14 heterodimer, competing with the substrate-binding site (SAM pocket). Reduces AML engraftment and prolongs survival in mouse models.	AML	[65]
R-2 hydroxyglutarate (FTO inhibitor)	Competitively occupying the α-KG site in FTO’s catalytic pocket. Raises global m6A levels on specific leukemic mRNAs, which leads to reduced MYC/CEBPA expression, curbing leukemic proliferation/survival. Inhibits aerobic glycolysis without affecting HSCs.	Sensitive leukemia cells	[54,66]
FB23 and FB23-2 (selective small molecules as FTO inhibitors)	Directly bind and inhibit FTO, thus increase m6A levels on key leukemogenic transcripts such as MYC/CEBPA, impairing the proliferative and survival programs of AML cells and inhibiting their progression.	AML	[67]
Glutathione (GSH) bioimprinted nanocomposite material, GNPIPP12MA, loaded with FTO inhibitors	GNPIPP12MA is engineered to bind/traffic in GSH-rich leukemic stem cells targeting the FTO/m6A pathway, synergistically. Enhances anti-leukemia effects by depleting intracellular GSH, which increases lipid peroxidation and iron-dependent damage, and drives selective ferroptosis.	Leukemia stem cells	[68]
Zantrene (FTO inhibitor and cytotoxic)	Dual mechanism of epitranscriptomic inhibition (by increasing m6A levels) plus traditional cytotoxicity (by causing single-strand DNA breaks and DNA–protein crosslinks).	AML	[69]
2-(2-hydroxyethylsulfanyl)acetic acid (RD3) and 4-[(methyl)amino]-3,6-dioxo (RD6) (small molecules acting as ALKBH5 inhibitors)	Both are most plausibly reversible active-site binders on ALKBH5 enzyme (metal-chelators or substrate pocket blockers rather than covalent modifiers). Decrease cell viability at low concentrations.	Leukemia cell lines	[70]
Pyrazolo [1,5-a]pyrimidine (DO-2728a) (small molecule acting as ALKBH5 inhibitor)	Enhances m6A modification abundance and impedes cell cycle progression.	AML	[71]
Covalent inhibitor TD19 (selective ALKBH5 inhibitor)	Irreversibly modifies specific cysteine residues (C100 and C267) in ALKBH5 enzyme near its active site.	AML	[72]

**Table 2 cimb-47-00758-t002:** Endo-cellular miRs contributing to MDS and AML hematologic neoplasias. The cell type source is bone marrow mononuclear cells, BMMCs. MiR level comparisons are against physiological controls or samples from patients with aplastic anemia (AA) [128].

MiRNA	Regulation in MDS/AML vs. Healthy or AA (Aplastic Anemia)
hsa-miR-130a-3p	Significantly higher in de novo MDS vs. AA
hsa-miR-221-3p	Significantly higher in de novo MDS vs. AA
hsa-miR-126-3p	Significantly higher in de novo MDS vs. AA
hsa-miR-27b-3p	Significantly higher in de novo MDS vs. AA
hsa-miR-196b-5p	Significantly higher in de novo MDS vs. AA
hsa-let-7e-5p	Upregulated in de novo MDS vs. AA
hsa-miR-181c-5p	Associated with progression to sAML
hsa-miR-155	Downregulated in MDS (prognostic value)

## Data Availability

No data were used for the research described in this article.

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
