# Peer review of "Intracellular Mis-Localization of Modified RNA Molecules and Non-Coding RNAs: Facts from Hematologic Malignancies"

_cimb, 2025, doi:10.3390/cimb47090758_

Round 1
Reviewer 1 Report
Comments and Suggestions for Authors
Symeonidis A. and coauthors, in the review entitled “Intracellular Mis-Localization of Modified RNA Molecules and 2 Non-Coding RNAs: Facts from Hematologic Malignancies”, discuss the intracellular wrong localization of modified RNA molecules and non-coding RNAs in the context of different hematological malignancies such as acute myeloid leukemia (AML), Diffuse large B-cell lymphoma (DLBCL), myelodysplastic syndrome (MDS), Aplastic Anemia (AA), multiple myeloma or Chronic lymphocytic leukemia compared to “physiological” conditions.
The authors describe how RNA modifications impact these diseases by influencing gene expression, RNA stability, and cellular processes. They discuss extensively about the most common chemical modifications of RNA such as n6-Methyladenosine (m6A), 5-Methylcytosine (m5C), Pseudouridine (Ψ), explaining their biological impact in both normal hematopoietic homeostasis and in pathological conditions.
In addition, the review provides relevance to ncRNAs, such as microRNA (miRs) and long non-coding RNAs (lncRNA), analyzing their intracellular distribution and their role in gene regulation and the pathogenesis of hematological malignancies. Indeed, mis-localization of miRs and lncRNAs can interfere with gene regulatory networks, contributing to hematologic malignancies. For example, lncRNA-HOTAIRM1 and lncRNA-HOTTIP are implicated in AML progression by altering chromatin structure and gene expression.
The review is well organized and systematically structured and includes an extensive list of references, demonstrating thorough literature research.
The Introduction section provides historical and scientific context, explaining the importance of epigenetic modifications and modified RNA in the field of hematological malignancies. Each section delves into molecular mechanisms, pathological implications, and potential therapeutic applications. The concluding section offers a synthesis of the evidence and suggests future directions for research and the development of RNA-based therapies.
However, the length and density of information may be difficult for some readers to understand.
This problem could be overcome by a more incisive summary or by a section with the key points of the topics.
Additional comments:
- Table 1 needs to be formatted because the contents of the columns overlap. Additionally, some points in the "Target" descriptions should be summarized more clearly.
- In fig 1 A It's not clear to me what the authors mean "deficient B cells" on the right of the cartoon.
- I would like to ask the authors to clarify when the results come from demonstrations obtained in murine models and possibly confirmed analogies in humans (e.g. lanes 152-156 refs. 28, 40-42); or when the cited results come from tumorigenic models other than hematological malignancies (e.g. lanes 187-189 ref. 47 refers to glioblastoma).
- Introduction section: lanes 52-53. “Since when Conrad Waddington…”in this context it would be useful to cite a reference (ie. Waddington C. H. (1942). The epigenotype. Endeavour, 18–20; and /or Waddington C. H. (1957). The Strategy of the Genes: A Discussion of Some Aspects of Theoretical Biology. London: Allen & Unwin) rather than the dates of birth and death (1905-1975)
- Introduction section: lanes 91-94 “Under physiological conditions NPM1 phosphoprotein shuttles…” the authors cite the reference 7 that is a recent correction of a paper published in 2008. The correct citation is the following: R. Garzon et al., Distinctive microRNA signature of acute myeloid leukemia bearing cytoplasmic mutated nucleophosmin. Proc. Natl. Acad. Sci. U.S.A. 105, 3945–3950 (2008). Correction in: Proc. Natl. Acad. Sci. U.S.A. 121, e2415544121 (2024), 10.1073/ pnas.2415544121.
Author Response
Reviewer 1
Comments and Suggestions for Authors
Symeonidis A. and coauthors, in the review entitled “Intracellular Mis-Localization of Modified RNA Molecules and 2 Non-Coding RNAs: Facts from Hematologic Malignancies”, discuss the intracellular wrong localization of modified RNA molecules and non-coding RNAs in the context of different hematological malignancies such as acute myeloid leukemia (AML), Diffuse large B-cell lymphoma (DLBCL), myelodysplastic syndrome (MDS), Aplastic Anemia (AA), multiple myeloma or Chronic lymphocytic leukemia compared to “physiological” conditions.
The authors describe how RNA modifications impact these diseases by influencing gene expression, RNA stability, and cellular processes. They discuss extensively about the most common chemical modifications of RNA such as n6-Methyladenosine (m6A), 5-Methylcytosine (m5C), Pseudouridine (Ψ), explaining their biological impact in both normal hematopoietic homeostasis and in pathological conditions.
In addition, the review provides relevance to ncRNAs, such as microRNA (miRs) and long non-coding RNAs (lncRNA), analyzing their intracellular distribution and their role in gene regulation and the pathogenesis of hematological malignancies. Indeed, mis-localization of miRs and lncRNAs can interfere with gene regulatory networks, contributing to hematologic malignancies. For example, lncRNA-HOTAIRM1 and lncRNA-HOTTIP are implicated in AML progression by altering chromatin structure and gene expression.
The review is well organized and systematically structured and includes an extensive list of references, demonstrating thorough literature research.
The Introduction section provides historical and scientific context, explaining the importance of epigenetic modifications and modified RNA in the field of hematological malignancies. Each section delves into molecular mechanisms, pathological implications, and potential therapeutic applications. The concluding section offers a synthesis of the evidence and suggests future directions for research and the development of RNA-based therapies.
However, the length and density of information may be difficult for some readers to understand.
This problem could be overcome by a more incisive summary or by a section with the key points of the topics.
Our response: We gratefully thank the reviewer for this recommendation, which multiplies the level of scientific accuracy and future readability of our manuscript. We have now added the key points of this review after the abstract section (lines 67-85).
Additional comments:
Table 1 needs to be formatted because the contents of the columns overlap. Additionally, some points in the "Target" descriptions should be summarized more clearly.
Our response: We thank the reviewer for his/her careful observations. We have reformatted table 1 and included detailed information within table lane, related to the drugs’ targets.
In fig 1 A It's not clear to me what the authors mean "deficient B cells" on the right of the cartoon.
Our response: We have now omitted this confusing phrase. Further to this we have split figure 1 into two separate figures, as we have also outlined the RNA modifications presented in the manuscript.
I would like to ask the authors to clarify when the results come from demonstrations obtained in murine models and possibly confirmed analogies in humans (e.g. lanes 152-156 refs. 28, 40-42); or when the cited results come from tumorigenic models other than hematological malignancies (e.g. lanes 187-189 ref. 47 refers to glioblastoma).
Our response: The study of RNA modifications in hematologic malignancies highlights the necessity for combining human-derived and animal-based disease models. Each of the two sources contributes unique strengths, and only through their integration emerging findings can achieve, both, mechanistic depth and translational relevance. We have included published data from human cell lines, primary human samples and mouse models, taking into consideration that complementary use of models ensures that discoveries are both, biologically rigorous and clinically relevant.
Cell lines serve as an accessible platform for molecular dissection. Their tractability allows for genetic perturbation, drug testing and mechanistic interrogation of pathways. A serious drawback is their long-term adaptation to culture and the lack of microenvironmental context, that both, limit their ability to fully recapitulate human disease complexity. Human cells provide the most authentic representation of the disease biology. They retain patient-specific mutational landscapes, epigenetic states, and RNA modification profiles, thereby ensuring that experimental observations are directly reflective of the real clinical situation. The major challenge lies in their scarcity, heterogeneity, and limited experimental manipulability, restricting large-scale mechanistic studies. Finally, mouse models, particularly genetically engineered strains and bone marrow transplantation systems, offer a powerful framework to establish causality. They enable the assessment of hematopoietic development, leukemogenesis, immune responses, and therapeutic intervention in the in vivo context. These models can reveal systemic consequences of molecular perturbations, that cannot be studied at the isolated human cell level. Nonetheless, individual differences in RNA metabolism and hematopoiesis necessitate careful validation of findings obtained from human material. In conclusion, we have included all studies related to this review topic, without commenting on disease models.
Publications mentioned (now Refs No 32, 44-46 in the revised clean version) have utilized combinations of patient samples, cell lines and mouse knockout or mouse bone marrow transplantation models, and we consider all of them relevant to the review purposes. Reference 47 in the revised clean version refers to glioblastoma, however, it introduces the readers globally to the general topic of our review and general discussion constitutes a large part of the manuscript, since we consider it a necessary contextual framework for the deeper understanding of this review subject.
Introduction section: lanes 52-53. “Since when Conrad Waddington…”in this context it would be useful to cite a reference (ie. Waddington C. H. (1942). The epigenotype. Endeavour, 18–20; and /or Waddington C. H. (1957). The Strategy of the Genes: A Discussion of Some Aspects of Theoretical Biology. London: Allen & Unwin) rather than the dates of birth and death (1905-1975).
Our response: Both of Waddington’s references have been included (New Refs No 1 and 2).
Introduction section: lanes 91-94 “Under physiological conditions NPM1 phosphoprotein shuttles…” the authors cite the reference 7 that is a recent correction of a paper published in 2008. The correct citation is the following: R. Garzon et al., Distinctive microRNA signature of acute myeloid leukemia bearing cytoplasmic mutated nucleophosmin. Proc. Natl. Acad. Sci. U.S.A. 105, 3945–3950 (2008). Correction in: Proc. Natl. Acad. Sci. U.S.A. 121, e2415544121 (2024), 10.1073/ pnas.2415544121.
Our response: The reference has been revised according to the reviewers’ very helpful information provided (now Ref No 11).
Reviewer 1
Comments and Suggestions for Authors
Symeonidis A. and coauthors, in the review entitled “Intracellular Mis-Localization of Modified RNA Molecules and 2 Non-Coding RNAs: Facts from Hematologic Malignancies”, discuss the intracellular wrong localization of modified RNA molecules and non-coding RNAs in the context of different hematological malignancies such as acute myeloid leukemia (AML), Diffuse large B-cell lymphoma (DLBCL), myelodysplastic syndrome (MDS), Aplastic Anemia (AA), multiple myeloma or Chronic lymphocytic leukemia compared to “physiological” conditions.
The authors describe how RNA modifications impact these diseases by influencing gene expression, RNA stability, and cellular processes. They discuss extensively about the most common chemical modifications of RNA such as n6-Methyladenosine (m6A), 5-Methylcytosine (m5C), Pseudouridine (Ψ), explaining their biological impact in both normal hematopoietic homeostasis and in pathological conditions.
In addition, the review provides relevance to ncRNAs, such as microRNA (miRs) and long non-coding RNAs (lncRNA), analyzing their intracellular distribution and their role in gene regulation and the pathogenesis of hematological malignancies. Indeed, mis-localization of miRs and lncRNAs can interfere with gene regulatory networks, contributing to hematologic malignancies. For example, lncRNA-HOTAIRM1 and lncRNA-HOTTIP are implicated in AML progression by altering chromatin structure and gene expression.
The review is well organized and systematically structured and includes an extensive list of references, demonstrating thorough literature research.
The Introduction section provides historical and scientific context, explaining the importance of epigenetic modifications and modified RNA in the field of hematological malignancies. Each section delves into molecular mechanisms, pathological implications, and potential therapeutic applications. The concluding section offers a synthesis of the evidence and suggests future directions for research and the development of RNA-based therapies.
However, the length and density of information may be difficult for some readers to understand.
This problem could be overcome by a more incisive summary or by a section with the key points of the topics.
Our response: We gratefully thank the reviewer for this recommendation, which multiplies the level of scientific accuracy and future readability of our manuscript. We have now added the key points of this review after the abstract section (lines 67-85).
Additional comments:
Table 1 needs to be formatted because the contents of the columns overlap. Additionally, some points in the "Target" descriptions should be summarized more clearly.
Our response: We thank the reviewer for his/her careful observations. We have reformatted table 1 and included detailed information within table lane, related to the drugs’ targets.
In fig 1 A It's not clear to me what the authors mean "deficient B cells" on the right of the cartoon.
Our response: We have now omitted this confusing phrase. Further to this we have split figure 1 into two separate figures, as we have also outlined the RNA modifications presented in the manuscript.
I would like to ask the authors to clarify when the results come from demonstrations obtained in murine models and possibly confirmed analogies in humans (e.g. lanes 152-156 refs. 28, 40-42); or when the cited results come from tumorigenic models other than hematological malignancies (e.g. lanes 187-189 ref. 47 refers to glioblastoma).
Our response: The study of RNA modifications in hematologic malignancies highlights the necessity for combining human-derived and animal-based disease models. Each of the two sources contributes unique strengths, and only through their integration emerging findings can achieve, both, mechanistic depth and translational relevance. We have included published data from human cell lines, primary human samples and mouse models, taking into consideration that complementary use of models ensures that discoveries are both, biologically rigorous and clinically relevant.
Cell lines serve as an accessible platform for molecular dissection. Their tractability allows for genetic perturbation, drug testing and mechanistic interrogation of pathways. A serious drawback is their long-term adaptation to culture and the lack of microenvironmental context, that both, limit their ability to fully recapitulate human disease complexity. Human cells provide the most authentic representation of the disease biology. They retain patient-specific mutational landscapes, epigenetic states, and RNA modification profiles, thereby ensuring that experimental observations are directly reflective of the real clinical situation. The major challenge lies in their scarcity, heterogeneity, and limited experimental manipulability, restricting large-scale mechanistic studies. Finally, mouse models, particularly genetically engineered strains and bone marrow transplantation systems, offer a powerful framework to establish causality. They enable the assessment of hematopoietic development, leukemogenesis, immune responses, and therapeutic intervention in the in vivo context. These models can reveal systemic consequences of molecular perturbations, that cannot be studied at the isolated human cell level. Nonetheless, individual differences in RNA metabolism and hematopoiesis necessitate careful validation of findings obtained from human material. In conclusion, we have included all studies related to this review topic, without commenting on disease models.
Publications mentioned (now Refs No 32, 44-46 in the revised clean version) have utilized combinations of patient samples, cell lines and mouse knockout or mouse bone marrow transplantation models, and we consider all of them relevant to the review purposes. Reference 47 in the revised clean version refers to glioblastoma, however, it introduces the readers globally to the general topic of our review and general discussion constitutes a large part of the manuscript, since we consider it a necessary contextual framework for the deeper understanding of this review subject.
Introduction section: lanes 52-53. “Since when Conrad Waddington…”in this context it would be useful to cite a reference (ie. Waddington C. H. (1942). The epigenotype. Endeavour, 18–20; and /or Waddington C. H. (1957). The Strategy of the Genes: A Discussion of Some Aspects of Theoretical Biology. London: Allen & Unwin) rather than the dates of birth and death (1905-1975).
Our response: Both of Waddington’s references have been included (New Refs No 1 and 2).
Introduction section: lanes 91-94 “Under physiological conditions NPM1 phosphoprotein shuttles…” the authors cite the reference 7 that is a recent correction of a paper published in 2008. The correct citation is the following: R. Garzon et al., Distinctive microRNA signature of acute myeloid leukemia bearing cytoplasmic mutated nucleophosmin. Proc. Natl. Acad. Sci. U.S.A. 105, 3945–3950 (2008). Correction in: Proc. Natl. Acad. Sci. U.S.A. 121, e2415544121 (2024), 10.1073/ pnas.2415544121.
Our response: The reference has been revised according to the reviewers’ very helpful information provided (now Ref No 11).
Reviewer 2 Report
Comments and Suggestions for Authors
The review submitted by Argiris Symeonidis et al to CIMB is devoted to the role of non-coding RNAs in progression of hematological malignancies
Some minor issues do not allow the reviewer to recommend the manuscript for publication
1) Change Fig. 1A, 1B, 1C into Fig. 1, Fig. 2 and Fig. 3 and move them after the mentions in the text.
2) Authors should consider adding Figures with m6A, m5C and Ψ, as the article type (review) and journal format (no page or figure number restrictions) allow this
3) Refs should be added to Table 2
4) In the Section 4, references to Figures and Table look very strange and unusual
Author Response
Reviewer 2
Comments and Suggestions for Authors
The review submitted by Argiris Symeonidis et al to CIMB is devoted to the role of non-coding RNAs in progression of hematological malignancies
Some minor issues do not allow the reviewer to recommend the manuscript for publication
1) Change Fig. 1A, 1B, 1C into Fig. 1, Fig. 2 and Fig. 3 and move them after the mentions in the text.
2) Authors should consider adding Figures with m6A, m5C and Ψ, as the article type (review) and journal format (no page or figure number restrictions) allow this
Our response for 1) and 2): We thank the reviewer for these constructive points. We have now added a new figure 1 including the RNA modifications discussed in our manuscript and have split previous figure 1(A, B and C), into 2 separate figures. However, we have kept illustrations referring both to AML in the same figure (2A and 2B) as they are strongly related.
3) Refs should be added to Table 2
Our response: References have been added.
4) In the Section 4, references to Figures and Table look very strange and unusual.
Our response: Within the text all relevant information has been provided, however, we have enriched this paragraph with additional text and a new reference. To avoid the concise (and perhaps undetailed) descriptions in the figure legends, we have decided to omit references.
Reviewer 3 Report
Comments and Suggestions for Authors
This review attempts to synthesize the complex interplay between RNA localization and hematologic malignancies. While the manuscript provides a reasonable overview of several RNA modifications and non-coding RNAs, its contribution is diminished by several critical flaws that require significant attention.
Here are the primary concerns with the manuscript in its current form:
-
Incomplete Scope and Omission of Key Epitranscriptomic Marks: The most significant weakness of this review is its failure to address the full spectrum of RNA modifications relevant to hematologic cancers. The complete omission of adenosine-to-inosine (A-to-I) editing is a major oversight. A-to-I editing, mediated by ADAR enzymes, is a critical regulator of hematopoietic stem cell function, and its dysregulation is extensively documented in the development and progression of various leukemias and lymphomas. By ignoring this entire area of research, the review presents a skewed and incomplete picture of the epitranscriptomic landscape in these diseases.
-
Lack of a Cohesive and Consistently Argued Central Theme: The review posits that the "mis-localization" of RNA is a central driver of hematologic malignancies. However, this theme is not consistently and robustly argued throughout the manuscript. While some examples, such as the subcellular trafficking of specific microRNAs, are directly relevant, many sections digress into a general discussion of the dysregulation of RNA-modifying "writers," "erasers," and "readers." In these instances, the connection back to a clear defect in RNA's subcellular compartmentalization is often tenuous or absent, leaving the reader to question the manuscript's core thesis.
-
Outdated and Incomplete Discussion of Therapeutic Landscape: The section on therapeutic interventions does not reflect the current state of the field. It overlooks several key advancements and emerging strategies that are critical for a contemporary review. For example, there is no mention of the development and entry into clinical trials of small molecule inhibitors targeting specific RNA-modifying enzymes, a crucial area of therapeutic research. This omission provides the reader with a dated perspective on the translational potential of targeting the epitranscriptome in hematologic cancers.
In summary, while the paper has a promising foundation, its significant omissions, lack of a consistently developed central argument, and outdated therapeutic summary prevent it from being a valuable contribution to the literature in its current state.
Author Response
Reviewer 3
Comments and Suggestions for Authors
This review attempts to synthesize the complex interplay between RNA localization and hematologic malignancies. While the manuscript provides a reasonable overview of several RNA modifications and non-coding RNAs, its contribution is diminished by several critical flaws that require significant attention.
Here are the primary concerns with the manuscript in its current form:
Incomplete Scope and Omission of Key Epitranscriptomic Marks: The most significant weakness of this review is its failure to address the full spectrum of RNA modifications relevant to hematologic cancers. The complete omission of adenosine-to-inosine (A-to-I) editing is a major oversight. A-to-I editing, mediated by ADAR enzymes, is a critical regulator of hematopoietic stem cell function, and its dysregulation is extensively documented in the development and progression of various leukemias and lymphomas. By ignoring this entire area of research, the review presents a skewed and incomplete picture of the epitranscriptomic landscape in these diseases.
Our response: We thank the reviewer for this important notification, which escaped from our focus. We have now added a new paragraph, related to the consequences deriving from mis-localization of this specific RNA editing (A→I) and responsible modifying ADAR enzymes, oriented to hematologic malignancies (lines 335-383, new Refs No 84-101).
Lack of a Cohesive and Consistently Argued Central Theme: The review posits that the "mis-localization" of RNA is a central driver of hematologic malignancies. However, this theme is not consistently and robustly argued throughout the manuscript. While some examples, such as the subcellular trafficking of specific microRNAs, are directly relevant, many sections digress into a general discussion of the dysregulation of RNA-modifying "writers," "erasers," and "readers." In these instances, the connection back to a clear defect in RNA's subcellular compartmentalization is often tenuous or absent, leaving the reader to question the manuscript's core thesis.
Our response: We sincerely thank the reviewer for this thoughtful observation. We agree that maintaining a clear and cohesive central theme is essential. Our intent in including broader discussions on RNA-modifying "writers," "erasers," and "readers" was to provide readers with a necessary contextual framework, especially for those less familiar with the field. We felt this general overview would help situate the more specific examples of RNA mis-localization within the wider landscape of RNA biology and with specific focus on hematologic malignancies.
Outdated and Incomplete Discussion of Therapeutic Landscape: The section on therapeutic interventions does not reflect the current state of the field. It overlooks several key advancements and emerging strategies that are critical for a contemporary review. For example, there is no mention of the development and entry into clinical trials of small molecule inhibitors targeting specific RNA-modifying enzymes, a crucial area of therapeutic research. This omission provides the reader with a dated perspective on the translational potential of targeting the epitranscriptome in hematologic cancers.
Our response: We thank you for these comments, aiming to further improve the informatory potential of our review. We would like to make clear that Table 1 refers exclusively to m6A RNA modification (the most prevalent and well-studied) and now presents also recent evidence of an ongoing clinical trial. FTO/ ALKBH5 inhibitors are still in preclinical stage however, table has been updated with extended information on drugs’ targets. Moreover, we have updated the reference list by incorporating several new, recently published articles (new refs No 65, 69, 71 and 72).
In summary, while the paper has a promising foundation, its significant omissions, lack of a consistently developed central argument, and outdated therapeutic summary prevent it from being a valuable contribution to the literature in its current state.
Our response: We hope that, by addressing all the comments and recommendations of the reviewers, we have substantially improved our manuscript (new refs in the “introduction” section No 5,6 and No 158-161 in the “Conclusions”). The new version will hopefully make you to change this opinion.
Round 2
Reviewer 3 Report
Comments and Suggestions for Authors
The manuscript is greatly improved and I recommend publication of this review.